# Muon $g-2$/EDM measurement at J-PARC

M. Abe[1], S. Bae[2, 3], G. Beer[4], G. Bunce[5], H. Choi[2, 3], S. Choi[2, 3], M. Chung[6], W. da Silva[7],
S. Eidelman[8, 9, 10], M. Finger[11], Y. Fukao[1], T. Fukuyama[12], S. Haciomeroglu[13],
K. Hasegawa[14], K. Hayasaka[15], N. Hayashizaki[16], H. Hisamatsu[1], T. Iijima[17], H. Iinuma[18],
K. Inami[17], H. Ikeda[19], M. Ikeno[1], K. Ishida[20], T. Itahashi[12], M. Iwasaki[20], Y. Iwashita[21],
Y. Iwata[22], R. Kadono[1], S. Kamal[23], T. Kamitani[1], S. Kanda[20], F. Kapusta[7], K. Kawagoe[24],
N. Kawamura[1], R. Kitamura[14], B. Kim[2, 3], Y. Kim[25], T. Kishishita[1], H. Ko[2, 3], T. Kohriki[1],
S. Kamioka[1], Y. Kondo[14], T. Kume[1], M. J. Lee[13], S. Lee[13], W. Lee[26], G. M. Marshall[27],
Y. Matsuda[28], T. Mibe[1, 29], Y. Miyake[1], T. Murakami[1], K. Nagamine[1], H. Nakayama[1],
S. Nishimura[1], D. Nomura[1], T. Ogitsu[1], S. Ohsawa[1], K. Oide[1], Y. Oishi[1], S. Okada[31],
A. Olin[4, 27], Z. Omarov[25], M. Otani[1], G. Razuvaev[8, 9 *], A. Rehman[29], N. Saito[1, 30],
N. F. Saito[20], K. Sasaki[1], O. Sasaki[1], N. Sato[1], Y. Sato[1], Y. K. Semertzidis[25], H. Sendai[1],
Y. Shatunov[31], K. Shimomura[1], M. Shoji[1], B. Shwartz[9, 31], P. Strasser[1], Y. Sue[17], T. Suehara[24],
C. Sung[6], K. Suzuki[17], T. Takatomi[1], M. Tanaka[1], J. Tojo[24], Y. Tsutsumi[24], T. Uchida[1],
K. Ueno[1], S. Wada[20], E. Won[26], H. Yamaguchi[1], T. Yamanaka[24], A. Yamamoto[1], T. Yamazaki[1],
H. Yasuda[28], M. Yoshida[1] and T. Yoshioka[24]

**1** High Energy Accelerator Research Organization (KEK), Ibaraki, Japan
**2** Seoul National University, Seoul, Republic of Korea
**3** Institute for Nuclear and Particle Astrophysics, Seoul, Republic of Korea
**4** University of Victoria, British Columbia, Canada
**5** Retired, Boulder, Colorado, USA
**6** UNIST, Ulsan, Republic of Korea
**7** LPNHE (CNRS/IN2P3/UPMC/UDD), Paris, France
**8** Budker Institute of Nuclear Physics, Novosibirsk, Russia
**9** Novosibirsk State University, Novosibirsk, Russia
**10** Lebedev Physical Institute RAS, Moscow, Russia
**11** Charles University, Prague, Czech Republic
**12** Osaka University, Osaka, Japan
**13** Institute for Basic Science (IBS), Daejeon, Republic of Korea
**14** Japan Atomic Energy Agency (JAEA), Ibaraki, Japan
**15** Niigata University, Niigata, Japan
**16** Tokyo Institute of Technology, Tokyo, Japan
**17** Nagoya University, Aichi, Japan
**18** Ibaraki University, Ibaraki, Japan
**19** Japan Aerospace Exploration Agency (JAXA), Tokyo, Japan
**20** RIKEN, Saitama, Japan
**21** Kyoto University, Kyoto, Japan
**22** National Institute of Radiological Sciences (NIRS), Chiba, Japan
**23** University of British Columbia, British Columbia, Canada
**24** Kyushu University, Fukuoka, Japan
**25** Korea Advanced Institute of Science and Technology (KAIST), Daejeon, Republic of Korea
**26** Korea University, Seoul, Republic of Korea
**27** TRIUMF, British Columbia, Canada
**28** The University of Tokyo, Tokyo, Japan
**29** Graduate University for Advanced Studies (SOKENDAI), Ibaraki, Japan
**30** J-PARC Center, Ibaraki, Japan
**31** Chubu University, Aichi, Japan
* g.p.razuvaev@inp.nsk.su

January 10, 2022

## Abstract

The muon $g-2$/EDM experiment at J-PARC is under preparation and targeted to measure the muon anomalous magnetic moment with the precision of 450 ppb and muon electric dipole moment with $1.5 \times 10^{-21}\,e$ cm at its first stage, thus contributing to investigation of discrepancy between the Standard Model prediction and the current world average of muon $g-2$. The latter is dominated by two similar experiments E821 BNL and E989 FNAL, while we suggest a novel approach: pulsed primary proton beam provides surface muons, which are diffused through a silica aerogel target forming thermalised muonium atoms. They are laser ionised and re-accelerated by a multi-stage linac up to 300 MeV/$c$ before spiral injection into the storage uniform 3 T MRI-like magnet volume at the stable orbit in the absence of E-field. The silicon strip detector placed inside the magnet measures decayed positron parameters used in data analysis. We report the experimental approach, current status, and future prospects.

## Contents

# 1 Introduction

The Standard Model (SM) is the main theoretical framework to interpret and predict phenomena in particle physics. Despite success of the SM in describing many observation, it is known that it is not complete missing gravitation, dark matter, dark energy, *etc.* Search for the so called new physics or physics beyond the SM is ongoing in numerous frontiers. One of them is a precision physics, when a search for the discrepancy between a measurement and its SM prediction is done with high accuracy. Good examples of such cases are measurements of muon properties such as magnetic dipole moment $\mu_\mu$ and electric dipole moment $d_\mu$, which gives the following contributions to the Hamiltonian:

$$\mathcal{H} = -\vec{\mu}_\mu \vec{H} - \vec{d}_\mu \vec{E}, \tag{1}$$

where $\vec{H}$ and $\vec{E}$ are magnetic and electric fields. $\mu_\mu$ and $d_\mu$ can be rewritten in another terms as

$$\vec{\mu}_\mu = g_\mu \frac{e}{2m} \vec{s}, \ \vec{d}_\mu = \eta_\mu \frac{e}{2mc} \vec{s}. \tag{2}$$

Here $g_\mu$ is gyromagnetic factor, $\eta_\mu$ — a factor for the EDM, $e$ and $m$ — particle's electric charge and mass. $g_\mu$ in the tree level diagram is equal to 2 and all radiation corrections are notated as the anomalous magnetic moment $a_\mu = (g_\mu - 2)/2$.

The precision of $a$ measurements and theoretical prediction are increasing with time, what summarised in pic. where a long standing deviation is seen. The nowadays experimental value is leaded by two experiments BNL E821 [1] and FNAL E989 [2]. The E821 published its final result in 2006, the E989 revealed result of the Run 1 in April 2021 and is going to collect the total statics 20 times higher than at E821. Both experiments rely on the use of so called "magic" momentum, allowing them the use of electric focusing, what in general constrains the accelerator part, so both results share some systematic error sources.

The independent method would be highly appreciated to cross check the current tension between experimental average and SM prediction [3] of $4.2\sigma$. Such novel approach is proposed in J-PARC (Japan proton accelerator research centre, Tokai-mura, Japan). New technique is rely on use of a low emittance muon beam stored in a high uniform magnetic field region without electric field focusing, what is expected to provide better systematic uncertainties and achieve at the Phase-I the same statistic precision as in E821.

Searches for permanent electric dipole moments (EDM) of fundamental particles are the experiments most sensitive to new *CP* violating physics. The most strong limit on value of a muon EDM $d_\mu$ have been set by the E821 experiment [4] to the $1.9 \times 10^{-19}$ $e$ cm at 90 % C. L. and in the new experiment we are going to achieve a ~50 times stronger limit.

The content of the paper is the following: firstly the experiments concept is reviewed, then main experimental components explained and their status is provided, while conclusion encloses the outlook.

# 2 Idea

The spin precision frequency around momentum in orthogonal *E*- and *B*-fields is described with the help of the Bargmann–Michel–Telegdi equation:

$$\vec{\omega} = \vec{\omega}_a + \vec{\omega}_\eta = -\frac{e}{m} \left[ a_\mu \vec{B} - \left( a_\mu - \frac{1}{\gamma^2 - 1} \right) \frac{\vec{\beta} \times \vec{E}}{c} + \frac{\eta_\mu}{2} \left( \vec{\beta} \times \vec{B} + \frac{\vec{E}}{c} \right) \right]. \tag{3}$$

While the "magic" momentum enforce the exclusion of the second term by requiring $a_\mu = 1/(\gamma^2 - 1)$, the absence of the electric field is suggested to leave only coupling of the spin to the magnetic

field through $a_\mu$ and $\eta_\mu$:

$$\vec{\omega} = \vec{\omega}_a + \vec{\omega}_\eta = -\frac{e}{m}\left[a_\mu\vec{B} + \frac{\eta_\mu}{2}\vec{\beta}\times\vec{B}\right].\qquad(4)$$

This lets one to choose a muon momenta to store particles in a MRI-like magnet with a highly uniform magnetic field, using a three-dimensional spiral injection providing a significantly better injection efficiency than in a kicker-inflector in-plane injection.

On other hand it would require the development of a low-emittance muon beam source.

# 3   Experiment

The proposed experiment described in [5] will take place at the Japan Proton Accelerator Research Center (J-PARC) at Tokai, Japan. J-PARC has a proton linac which produces a 400 MeV H$^-$ beam with 50 mA peak current and ∼350 μs pulse width. The negative hydrogen ions are injected into the 3 GeV rapid cycling synchrotron and then moved through the channel to the Material and Life-science Facility (MLF) with the 25 Hz repetition rate. The MLF has a 2 cm thick carbon target to produce a surface muon beam from pions decayed near and at the surface of the target. The target is surrounded by capturing magnets of four muon lines. The last line being constructed is the H-line.

## 3.1   H-line

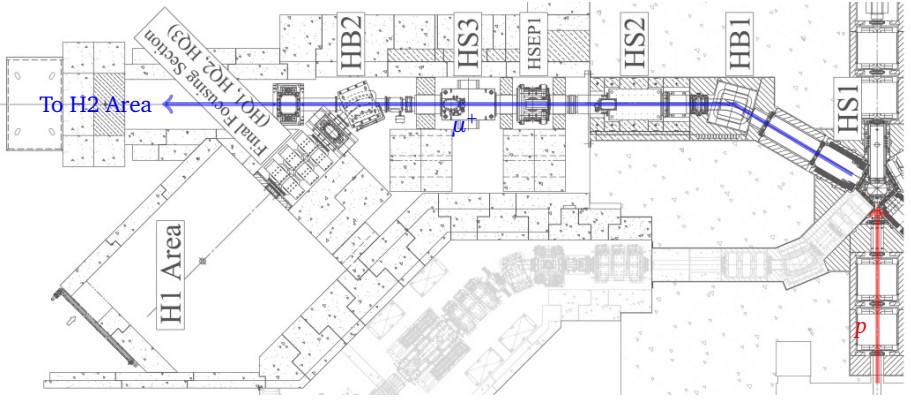

Figure 1: The H-line layout. (Adopted from [6].)

The H-line is a high intensity surface muon beamline [6] for long time experiments producing $1.6\times10^8$ 100 % polarised $\mu^+$/s at 1 MW proton beam power. The beamline layout is presented in fig. 1. The H-line consists of a wide angle capturing solenoid, bending magnets, focusing solenoids, and a separator. The line are split into two channels by a bending magnet HB2 to deliver the beam to the H1-area either to the H2-area. The H1-channel has three quadruple magnets to provide a beam with desirable parameters for such an experiment as DeeMe [7] or MuSEUM [8] in near future. The H2-channel is supposed to be used for the E34 experiment, and then for a muon microscope.

The current status are the following. The beamline are constructed to work for the H1-area and waits for the approval by Nuclear Regulation Authority in the beginning of FY 2022.

The H2-channel is under development. It should be elongated by a muon thermalisation device and a linac to prepare a low-emmitance beam for the $g-2$/EDM experiment. All that requires construction of an extension building to accommodate the linac, injection system,

150  storage magnet, control room, *etc.* The building is designed, the construction area is prepared,
151  completion of the extension building is scheduled in FY 2024.

## 3.2  Muon thermalisation

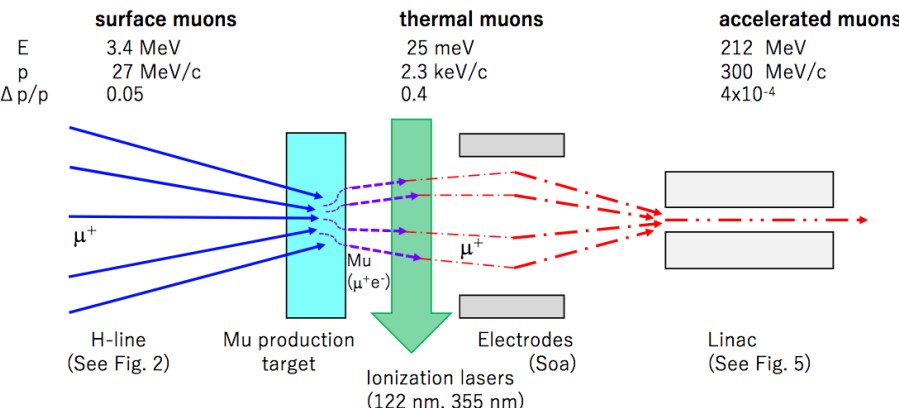

Figure 2: The muon thermalisation scheme. (Source [5].)

153  To give a base to the key feature of the experiment — no $E$-field at the storage region — the
154  low-emittance beam is of high priority. To get it the surface muon beam should be cooled with
155  a primary aim of reducing a transverse momentum spread: $\Delta p/p = 0.05$ for the surface $\mu^+$
156  beam to $4 \times 10^{-4}$ for re-accelerated muons after cooling. The muon cooling scheme is shown
157  in fig. 2, muons hit on the aerogel target, where they stop and some of them form neutral
158  muonium atoms (hydrogen-like $e^-$–$\mu^+$ bound state). A part of muoniums diffused out the
159  target and ionised by lasers. Thus thermalised muons are produced.

### 3.2.1  Muonium production

161  The surface muon beam is focused on the silica aerogel target, where muons stop and cap-
162  ture electrons to form muoniums. To increase the muonium diffusion rate the ablation of the
163  aerogel plates is used, [9].
164  The optimisation of ablation pattern have been done and its result that a double-side ab-
165  lated aerogel plate with primary density $\approx$23.6 mg/cm$^3$ with holes of $\sim$2 mm depth, 100 μm
166  to 250 μm diameter and the opening fraction of the ablation region around 0.6 gives the best
167  achieved diffusion rate, about 10 times increase comparing to a plane aerogel plate. The
168  aerogel samples good time stability was checked on the time period up to 2 days.
169  The target holder and the vacuum chamber to accommodate the holder and provide en-
170  trances for laser beams is under design.
171  The output muonium production efficiency is estimated to be 3.4 ‰, what fulfils design
172  requirements of the Phase-I of the experiment, while the Phase-II requires 3 times fold increase
173  of the total cooled muon beam. This motivates the ongoing development of new target schemes
174  like multi-layer design or focusing of diffused Mu.

### 3.2.2  Muonium ionisation

176  The muonium ground state energy is 13.6 eV. To overcome this strong bindings resonance
177  multi-photon ionisation are implied: excitation of Mu and then ionisation. It requires two
178  different laser systems. Two schemes are proposed.
179  The first scheme is using Lyman-$\alpha$ 122.09 nm pulse laser to make the dipole 1S–2P transi-
180  tion, [10]. The coherent Lyman-$\alpha$ light is generated by two-photon resonant four-wave mixing

in Kr gas pumped by pulsed lasers at 212.556 nm and 820.65 nm. The achieved power by using this scheme is 3 μJ/pulse.

The further power increasing to meet the project value of 100 μJ/pulse is focused on developing the pump laser beams power, exactly the larger crystal for the 212.556 nm amplification is needed.

The second ionisation scheme is uses already available technologies with a high-intensity 244 nm laser to get a good 2-photon M2 excitation efficiency within the S1–S2 transition. An experiment to validate such scheme and measure the Mu ionisation efficiency and improving precise Mu 1S–2S energy determination is proposed at J-PARC at the S-line [11], which aims to measure the muonium 1S–2S frequency $\Delta\nu_{S1-S2}$ and the mass ratio $m_\mu/m_e$. A slow muon test beamline has been assembled at the MLF S-line (S2 area), and the surface muon beam was checked at the MLF S-line in autumn 2021. In January 2022 the beamline will be used to conduct the experiment with an aerogel target and a 244 nm laser on the muonium 1S–2S excitation and ionisation. Using an 1S–2S scheme improves the cooled muon polarisation from 50 % to 2/3, [12].

In both cases, after the primary excitation of Mu, the ionisation with 3.4 eV work-out is done by 355 nm laser.

## 3.3 Re-acceleration

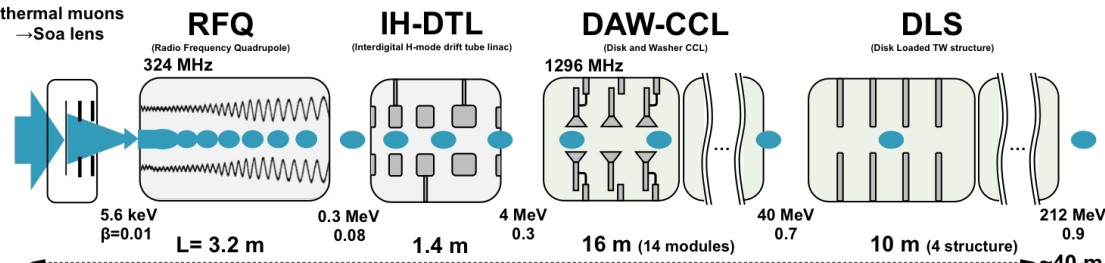

Figure 3: The muon linac scheme.

Cooled muons get a rapid acceleration in the way to minimise decay loss and emittance growth. The linac scheme is revealed in fig. 3, where acceleration starts from collecting muons with an electrostatic SOA lens downstreaming them to a radio-frequency quadrupole (RFQ). The RFQ forms three bunches and accelerate them to the energy of 0.34 MeV. Bunches go to an interdigital H-type drift tube linac (IH-DTL), then to the coupled-cavity linac with a disk-and-washer structure (DAW-CCL), and finally to a disk-loaded travelling wave structure (DLS). After that 212 MeV beam with momentum spread of 0.04 % (RMS) is ready for injection.

The SOA lens and a shorter RFQ prototype have been successfully tested in 2018 [13] and 2019 [14]. An RFQ originally produced for the J-PARC linac will be used for the $g-2$/EDM experiment [15]. The RFQ successfully passed an electrical test. The muon test beam with the RFQ at the H2-line is planned in 2022. An IH-DTL prototype, 3 times shorter then the project, is produced and have passed a low power test in 2019 [16]. The production of the full IH-DTL is planned by the end of the 2021 fiscal year. Basic DAW-CCL design is finished, the production of the first DAW-CCL tank is planned by the end of the 2021 FY [17]. The DLS is on final staged of detailed design [18].

## 3.4 Injection and storage

A MRI-like superconducting magnet is used for the storage of muons, see fig. 4. This technology provides a 3 T axial magnetic field with peak to peak 0.1 ppm local uniformity in the

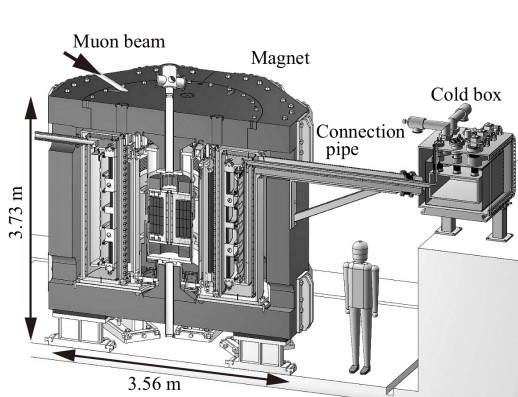
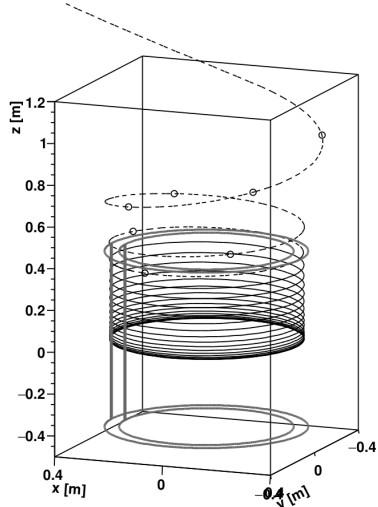

Figure 4: Overview of the storage magnet. (Source [5].)

Figure 5: A three-dimensional concept view of the beam trajectories from the injection (dashed line) through kicker region (solid line) to the storage. (Source [5].)

333 mm radius storage orbit, [19].

A three-dimensional spiral injection is chosen to deliver muons from the linac through the magnet top to a storage region. The beam from the linac output is inclined by 26° and injected into the magnet, then vertical motion is compensated by a pulse magnetic field kick, which stops muons in the storage region during their several revolutions (fig. 5).

The 3D-spiral injection scheme was demonstrated with continuous strongly $XY$-coupled electron beam [20]. Now the prototype is being upgraded to work with a bunched beam [21] and the magnetic kicker have been tested [22] and needs further studies.

The magnet design is almost finished, currently the work is going on coil structure optimisation and the magnet shimming is mastering with the MuSEUM's MRI-like magnet.

### 3.4.1  Field measurement

The injection region is measured with help of Hall probes with accuracy about 100 ppm.

Fixed water nuclear magnetic resonance (NMR) probes, situated near the storage region, is used to monitor the magnetic field with precision ∼0.05 ppm during data tacking, while mapping 0.01 ppm accurate water NMR probes will be used regular to get a 3D magnetic map in the storage region.

New NMR probes with $^3$He are under development, which promises better accuracy due to smaller correction than water probes.

### 3.5  Tracker

A silicon strip track detector, presented at fig. 6, is placed inside the storage orbit. The tracker consists of 40 plane modules situated radially. Each module has upper and lower parts. Such part has silicon strips on both sides. Strips on one side are vertical and horizontal on another. At each side there are four sensors, 1024 strips each. This results into 655 360 channels.

Each 128 strips are read out by an application-specific integrated circuit (ASIC). 8 ASICs are placed on one board, which is connected to an FRBS board. The board is supplied with a low power DC-DC converter providing a low disturbance of magnetic and electric fields [23].

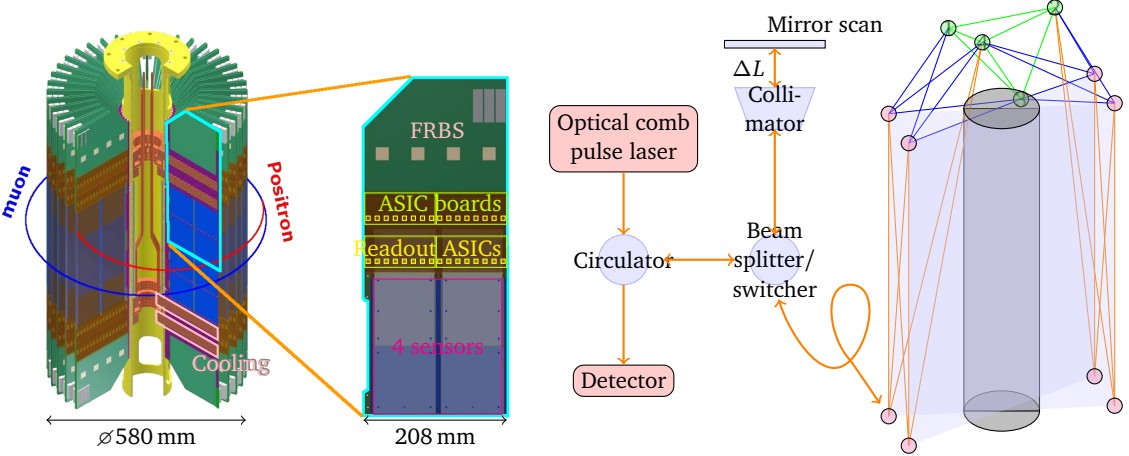

Figure 6: The positron tracker overview.

Figure 7: The alignment system principal scheme.

The signal from strips can be process with a differential or integration approach and then amplitude digitised with 5 ns sampling rate.

The estimated data taking rate from the detector is 360 MB/s.

The large part of electronics is ready for production. Four prototype modules were produced and successfully tested in MuSEUM runs at the MLF D2-line. One prototype module is used in the electron on proton scattering experiment ULQ2 [24].

## 3.6  Detector alignment

One of the key elements of proper EDM measurement is knowing the position of Si strip detectors during data taking cycle with accuracy $\leq 1\,\mu m$. Several steps are prescribed to achieve this.

Sensor positioning on the board is controlling with accuracy of $1\,\mu m$ during production.

Alignment/deformation monitor based on 3D-length measurement grid of absolute distance interferometers is under development to control position of 160 points, the concept is shown in fig. 7. A 2-point prototype confirmed required accuracy parameters.

A procedure to measure and control relative position of sensors using positron tracks themselves are under development.

## 3.7  Analysis and software

Full simulation of the experiment is divided in several parts: muon production at the target, beam conducting through H-line up to the aerogel target, thermalised muon production, re-acceleration, injection, muon storage and decay positron detection. Output of one step serves as an input for the subsequent part. Results of tracker response is used for analysis development and let one study various systematic errors in measurement of $a_\mu$ and $d_\mu$. The package for the detector simulation and positron track reconstruction is called g2esoft.

An additional package [25] has been develop to study systematic errors caused in later stages of the experiment as a pile-up effect, non-homogeneity of magnetic field, high energy positrons which can travel outside the detector volume.

Analysis starts with positron track finding and reconstruction. Track finding could be done by the usage Hough transformation or alternatively by using a multivariable analysis with boosted decision trees. Reconstructed positrons with high energy (200 MeV $< E <$ 275 MeV)

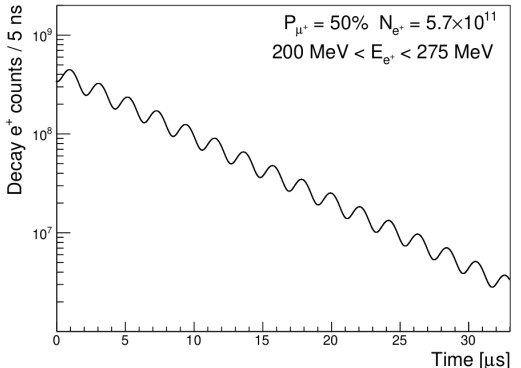
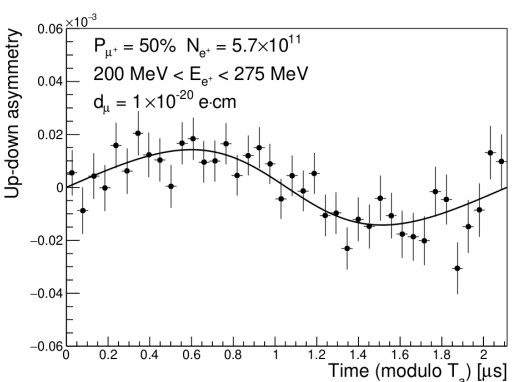

Figure 8: Simulated time distribution of reconstructed positrons. The solid curve is the fit to simulated data.

Figure 9: The simulated up-down asymmetry as a function of time modulo of the $g_\mu - 2$ period. The solid curve is the fit to simulated data.

are studied for a positron rate dependence on time to reveal an $\omega_a$ oscillation pattern, while a up-down asymmetry between positrons decay direction can shed a light on non-zero $d_\mu$.

# 4   Conclusion

The muon $g - 2$/EDM measurement at J-PARC is under preparation. Many its part were developed and are under production, while some are still under design or validation. The data taking is planned to start in 2026 with aim to achieve within the Phase-I the statistical and systematic precision on $\omega_a$ 450 ppb and 70 ppb respectively, while to set an upper limit on $d_\mu$ with $1.5 \times 10^{-21}\,e$ cm statistical and $0.36 \times 10^{-21}\,e$ cm systematical precision.

# Acknowledgements

The authors would like to thank the KEK and the J-PARC muon section staffs for their strong support.

**Funding information**   This work is supported by JSPS KAKENHI Grants No. JP19740158, No. JP23108001, No. JP23740216, No. JP25800164, No. JP26287053, No. JP26287055, No. JP15H03666, No. JP15H05742, No. JP16H03987, No. JP16J07784, No. JP16K13810, No. JP16K05323, No. JP17H01133, No. JP17H02904, No. JP17K05466, No. JP17K18784, No. JP18H01239 and No. JP18H03707. This work is also supported by the Korean National Research Foundation Grants No. NRF-2015H1A2A1030275, No. NRF-2015K2A2A4000092, and No. NRF-2017R1A2B3007018; the Russian Foundation for Basic Research Grant No. RFBR 17-52-50064 which is a part of the Japan–Russia Research Cooperative Program; the Russian Science Foundation Grant No. 17-12-01036; the Russian Ministry of Science and Higher Education Agreement 14.W03.31.0026; the U.S.–Japan Science and Technology Cooperation Program in High Energy Physics; the Discovery Grants Program of the Natural Sciences and Engineering Research Council of Canada; and the Institute for Basic Science (IBS) of Republic of Korea under Project No. IBS-R017-D1-2018-a00.

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
