# Peer review of "Muon g−2/EDM measurement at J-PARC"

_SciPost Physics Proceedings_

## Round 1 · Referee Report · Anonymous (Referee 1) · 2022-3-31

Strengths

1) a nice update of the status of an important experiment

Weaknesses

1) a bit unbalanced in details, but perhaps focusing more on what is now well understood vs things still in design optimization stage
2) with statements about improved systematics compared to BNL and FNAL approach, it would be natural to have a table of the systematics investigated so far and some reason to understand why one can achieve the goal
3) similarly the statistics limitation was not presented in terms of simple things like stored muons, fill rates, acceptance, asymmetry, etc. even though I am sure they have done that.

Report

This paper reads like a conference proceeding might; it is an update on a project that is better documented in topic-specific technical papers or a design report but it a nice and easy read and gave me a good feeling of where this project is and when they hope to take data so I believe it should be published.

Requested changes

Honestly, there are so many grammar issues I marked up far too many just on page 1 to even list. So, it needs a native English speaker or expert to go through and fix the obvious ones (it will be quite easy to do so)

Line 98 refers to a 'pic' but I didn't find it.
I'd like to see the tables of systematics and statistics if there is room

If there is time, another pass through to balance the sections would be valuable and to think about the level of details presented

  • validity: good
  • significance: good
  • originality: ok
  • clarity: ok
  • formatting: reasonable
  • grammar: below threshold

---

## Editorial Decision

in_voting